# A Community-Based, Participatory, Multi-Component Intervention Increased Sales of Healthy Foods in Local Supermarkets—The Health and Local Community Project (SoL)

**DOI:** 10.3390/ijerph20032478

**Published:** 2023-01-30

**Authors:** Ulla Toft, Tine Buch-Andersen, Paul Bloch, Helene Christine Reinbach, Bjarne Bruun Jensen, Bent Egberg Mikkelsen, Jens Aagaard-Hansen, Charlotte Glümer

**Affiliations:** 1Center for Clinical Research and Prevention, Bispebjerg and Frederiksberg Hospital, Nordre Fasanvej 57, 2000 Frederiksberg, Denmark; 2Department of Public Health, Section of Social Medicine, University of Copenhagen, Øster Farimagsgade 5, 1353 Copenhagen, Denmark; 3Health Promotion Research, Steno Diabetes Center Copenhagen, Borgmester Ib Juuls Vej 83, 2730 Herlev, Denmark; 4Department of Food Science, Section for Food Design and Consumer Behaviour, University of Copenhagen, Rolighedsvej 30, Building 2-74, 5th Floor, Room C505, 1958 Frederiksberg C, Denmark; 5Department of Geosciences and Natural Resource Management, University of Copenhagen, Rolighedsvej 23, 1958 Frederiksberg C, Denmark; 6SA MRC Developmental Pathways for Health Research Unit, Faculty of Health Sciences, University of the Witwatersrand, Johannesburg 2000, South Africa; 7Center for Diabetes, Copenhagen Municipality, Vesterbrogade 121, 1620 Copenhagen, Denmark

**Keywords:** multi-component intervention, mass media intervention, supermarkets, food sales, supersetting approach

## Abstract

Project SoL was a 19-month (September 2012 to April 2014) community-based multi-component intervention based on the supersetting approach that was designed to promote healthier eating and physical activity among children and their families. The aim of this study was to examine the effects of a multi-component intervention (level 1) and a mass media intervention alone (level 2) compared to a control area (level 3) on food sales. The design was quasi-experimental. Weekly sales data for all Coop supermarkets in the intervention and control areas were analysed via longitudinal linear mixed-effects analyses. Significant increases in the sales of fish (total) (29%; *p* = 0.003), canned fish (31%; *p* = 0.025) and oatmeal (31%; *p* = 0.003) were found for the level 1 intervention area compared to the control area. In the level 2 intervention area, significant increases in the sales of vegetables (total) (17%; *p* = 0.038), fresh vegetables (20%; *p* = 0.01), dried fruit (51%; *p* = 0.022), oatmeal (19%; *p* = 0.008) and wholegrain pasta (58%; *p* = 0.0007) were found compared to the control area. The sales of canned fish increased by 30% in the level 1 area compared to the level 2 area (*p* = 0.025). This study demonstrated significant increases in the sales of healthy foods, both in the areas with multi-component and mass media interventions alone compared to the control area.

## 1. Introduction

Substantial evidence has shown that healthy diets can promote better health and reduce the risk of obesity, non-communicable diseases (NDCs) and early death among adults [1]. Not least among children, good nutrition is essential for them to reach their growth and developmental potential [2]. Furthermore, dietary habits established during childhood tend to track into adulthood [3,4] and unhealthy diets among children have been found to increase the risk of becoming overweight and obese, as well as increasing the risk of NDCs later in life [1].

According to national dietary surveys, Danish children consume an inadequate amount of vegetables, wholegrains and fish and too much added sugar and total fat [5]. For example, only 6% and 23% of Danish 4–18-year-old children reach recommended intakes of fish and fruit and vegetables, respectively, whereas 50% of children consume more added sugar than the recommended <10 energy% [5,6,7]. However, promoting long-term changes in dietary habits is not an easy task. Hence, children and families’ food choices are influenced by a complex interplay between many factors in a complex system, including the availability of healthy food items in supermarkets, canteens and institutions; knowledge and awareness related to healthy eating, social and cultural norms; pricing and marketing [8]. Single-component approaches targeting individuals to promote healthier behaviour have only shown limited and short-term effects [9,10]. Hence, current evidence points to the need for long-term community-based and whole-system approaches that go beyond traditional information campaigns and target multiple settings where children and families spend their time and address a wide range of factors, including environmental influences within communities [11,12,13,14,15,16]. Furthermore, according to many studies and reviews, co-creation and involvement of relevant partners and parents increase the likelihood for success [14,15,17,18,19,20].

Project SoL, from the Danish ‘Sundhed og Lokalsamfund’ (Health and Local Community), aimed to mobilise local community resources, strengthen local social networks and promote physical activity, healthier eating and shopping behaviour among Danish children aged 3–8 years and their families living in selected communities on the Danish Island of Bornholm. The intervention was based on the supersetting approach, which implies that the intervention was implemented in a coordinated and integrated manner in multiple local community settings and involved multiple stakeholders to promote intensity, impact, synergy and sustainability [21]. The supersetting approach was developed based on the settings approach, which was introduced in the Ottawa Charter [22] and focuses on intervening in the places/settings where citizens live their daily lives and to intervene in a synchronised way that creates intensity and stronger effects by ‘doing more things in more places’. Hence, Project SoL intervened in schools, childcare centres and supermarkets, as well as the local mass media and social media. The multi-component intervention consisted of a broad range of activities, focusing on healthy food, physical activity and well-being. These included active play for children and families, tasting and cooking with children, healthy recipes, space management in supermarkets to promote healthier shopping, healthy lunch bag workshops, breakfast clubs, outdoor nature-based activities, mass media intervention focusing on healthy eating, etc. (more detail on the supersetting approach and Project SoL can be found in the Methods Section). The design and intervention of Project SoL was described in detail previously [23,24]. 

The purpose of the present study was to evaluate the effects of the multi-component intervention of Project SoL on food sales in supermarkets located within three intervention communities compared to a mass media intervention only and to a control area with no intervention. Hence, in this study, we compare three levels of intervention (Figure 1): level 1: multi-component intervention (three local communities on the island of Bornholm (SoL communities)); level 2: mass media intervention only (the rest of the island of Bornholm); level 3: control area (Odsherred Municipality, no intervention).

## 2. Materials and Methods

### 2.1. Design and Intervention

Project SoL was carried out in Denmark in two municipalities during a four-year period from 2012 to 2015, with a 19-month intervention period (September 2012 to April 2014). The project was designed as a quasi-experimental study with a matched pair of interventions (Bornholm) and a control (Odsherred) area. Bornholm is an island with a land mass of 588 square kilometres and a population of approximately 42,000 inhabitants. Odsherred municipality covers an area of 355 square kilometres, with about 32,500 inhabitants. Bornholm and Odsherred were purposively selected as the first level of sampling due to their similar sociodemographic characteristics, including high proportions of citizens with a low socio-economic position and high prevalence rates of health risk factors and NCDs [25,26]. Table 1 provides an overview of important characteristics of the two municipalities. 

A multi-component intervention (level 1) was applied in three selected local communities. This was a setting-based intervention and included local childcare centres, primary schools, supermarkets, local mass media and the local communities in general. The intervention components included participatory learning, change and design methods (Future Workshop Scenario [27], Mosaic Method [28]), educational activities, mass communication through mass and social media, brochures, recipes and posters, educational activities for children and their families and professionals in childcare centres, schools and supermarkets. Furthermore, the intervention included environmental or structural changes in schools, childcare centres and supermarkets, in addition to different local community activities and public events. Mental and social well-being were especially promoted by activities, such as outdoor activities in nature, establishing aesthetic environments, tasting good food, exploring the senses, having fun and building strong social networks [23]. Generally, the activities were aimed at promoting health by increasing knowledge, awareness, action competences, participation, integration and social cohesion, as well as modifying the overall policies and structures to ensure that they supported healthier choices and, thereby, contributed to reducing social inequality. Regarding healthy eating, the main focus was on increasing the intake of vegetables, fruits, fish and wholegrains, and on decreasing the intake of soft drinks, cakes and sweets. 

To ensure coordination and integration of activities across settings and communities, common overall themes were selected and addressed based on the ideas and interests of the local partners and by the focus of the project. Hence, although all the activities were implemented according to the overall principles in supermarkets, childcare centres and/or schools and mass media, they varied from setting to setting and also from community to community, depending on the priorities of the local stakeholders. Overall themes included ‘taste and senses’, ‘active play’, ‘visibility’, ‘fruit and vegetables’, ‘whole grains’, ‘fish’, ‘nature and movement’, ‘nature as a pantry’ and ‘healthy alternatives’. Interventions were not predetermined but developed jointly with citizens and professional stakeholders as the project unfolded. The intervention has been described in detail elsewhere [23,24].

The local schools and childcare centres in each of the three local communities were all included. 

Eight supermarkets of different sizes and from three different food retail groups were located in the three local intervention communities. They were all invited to participate in the project. One small independent grocery store chose not to participate. The other seven supermarkets, including one large, two medium and two small conventional supermarkets and two medium-sized discount supermarkets, participated actively throughout the study period. The supermarket chains involved in the intervention included Netto, Spar and Coop.

The mass media intervention was delivered especially by the local TV station, but also by local radio, newspapers and Facebook. The local TV2 Bornholm station systematically covered the major events and components of Project SoL broadcasted to the whole island of Bornholm. In addition, TV2 Bornholm developed a special programme for the study called ‘Just a little healthier’, which focused on inspiration for healthy eating and cooking. Furthermore, local newspapers and radio programmes also covered the activities of Project SoL.

The rest of Bornholm was only exposed to this mass media intervention (level 2) and, therefore, the design provides the opportunity to investigate the effect of different levels of intensity of the intervention. This is illustrated in Figure 1.

### 2.2. Data Collection Methods

Weekly sales data for all products sold in the period from January 2013 to June 2014 were delivered by Coop (the largest supermarket chain in Denmark) for all Coop supermarkets on Bornholm, both the in the SoL communities (level 1) and the rest of Bornholm (level 2) and in Odsherred (control area). Due to a shift in the data system in one of the supermarket chains (Netto) during the intervention period and difficulties in obtaining digital data from a small supermarket (Spar), only data from the Coop supermarket chain are included in this study. Unfortunately, due to a standard procedure in Coop, data from 2012 were erroneously deleted and, hence, data from the first four months of the intervention were unavailable.

Based on the main focus of the dietary intervention (increasing intakes of vegetables, fruits, fish and wholegrains, and on decreasing intakes of soft drinks, cakes and sweets) we included sales data for all kinds of fruits and vegetables (including fresh, frozen and dried), fish (fresh, frozen, canned), wholegrain cereals and pasta, sweets, sugary beverages, cakes and unhealthy snacks. The weight of the foods within each of these food categories was determined. Weight and number sold were used to calculate the total weight sold per week of each food category. Monthly sales were calculated based on weekly sales.

Index numbers were calculated for each month as: (weight of sold products n month ×/weight of sold products in January 2013) × 100. For example, an index number of 134 reflects a 34% increase in the amount of fresh vegetables sold in a month compared to the first month of 2013.

### 2.3. Statistical Analyses

Statistical analyses were performed using SAS statistical software v.9.4. We fitted a longitudinal linear mixed-effects model with the logged index numbers regressed on a time-dependent intervention variable with categories ‘SoL’ (multi-component intervention communities (level 1) *n* = 4), ‘Bornholm’ (the whole island of Bornholm except the SoL supermarkets, mass media intervention only (level 2), *n* = 12) and ‘control’ (Odsherred, no intervention (level 3), *n* = 11).

We used a random effect for supermarket to allow for heterogeneity among supermarkets, and an autoregressive AR1 correlation structure to account for larger similarities of observations closer in time on the same supermarket. We, furthermore, took into account the seasonal variation in sales during public holidays, as this variation might differ between local areas. For small intervention effects, the intervention effect on the logged index scale can be approximately interpreted as the proportional change in index number during the intervention period. The analysis was performed using the proc mixed procedure in SAS. Effects of the intervention were estimated by comparing the development from the first month (January 2013) to the average sale during the last three months (April, May and June 2014) of the intervention in the SoL supermarkets (level 1) and other supermarkets on Bornholm (level 2) compared to the control area (level 3).

## 3. Results

Results comparing the development in food sales in the intervention Coop supermarkets (level 1), other Coop supermarkets on Bornholm (level 2) and Coop supermarkets in the control area (level 3) are shown in Table 2. The results showed a 29% increase in the sale of fish (total) in the SoL supermarkets (level 1) compared to the control area (level 3) (*p* = 0.003). The sales of canned fish in the SoL supermarkets (level 1) increased by 31% compared to the control area (level 3) (*p* = 0.025) and by 30% compared to the rest of Bornholm (level 2) (*p* = 0.025). Furthermore, the SoL supermarkets (level 1) increased the sale of oatmeal by 31% (*p* = 0.003) and there was a borderline significant increase in the sale of vegetables (22%; *p* = 0.06) and dried fruit (60%; *p* = 0.06) compared to the control area (level 3).

The supermarkets on Bornholm (level 2) significantly increased the sale of vegetables in total by 17% (*p* = 0.038) and the sale of fresh vegetables by 20% (*p* = 0.01) compared to the control area (level 3). Furthermore, significant increases in the sales of dried fruit (51%; *p* = 0.022), oatmeal (19%; *p* = 0.008) and wholegrain pasta (58%; *p* = 0.0007) were found on Bornholm (level 2) compared to the control area (level 3).

No significant changes in the sales of sugary beverages, sweets, cakes and unhealthy snacks were found in the intervention supermarkets (levels 1 and 2) compared to the control area (level 3).

Figure 2, Figure 3, Figure 4 and Figure 5 illustrate the development in monthly sales from January 2013 to June 2014 for fish (total), canned fish, vegetables (total) and oatmeal, respectively (as examples). In general, the figures show a greater increase in the intervention supermarkets (level 1) during the intervention period compared to the control supermarkets (level 3). The sales varied during the intervention period in all supermarkets. In particular, a peak in sales was seen in July 2013 due to the tourist season. Furthermore, the development in sales differed between food groups. However, in general, the differences between the three intervention levels increased during the intervention period.

## 4. Discussion

The results showed a significant increase in the sales of fish (total and canned) and oatmeal and a borderline significant increase in the sales of vegetables (total) and dried fruit in the SoL supermarkets (level 1) compared to control supermarkets (level 3). Furthermore, a significantly greater increase in the sale of canned fish was found in the SoL supermarkets (level 1) compared to all other Coop supermarkets on Bornholm (level 2). Finally, a significantly greater increase in the sales of vegetables (total and fresh), dried fruit and wholegrain products (oatmeal and wholegrain pasta) and a borderline higher increase in the sales of fish (total) were found in areas with low-intensity intervention (level 2) compared to the control area (level 3). No significant effects were found on sales of fresh fruit, sweets and sugary beverages. In short, the multi-component intervention (level 1) showed the greatest effect on intake of healthy food items, but there was also some effect in the low-intensity (level 2) areas, as compared to the control (level 3). In contrast, there were no significant effects on reductions in unhealthy food items in level 1 or level 2 communities.

The results from this study are in line with several previous studies [29,30,31,32,33,34,35], showing significant beneficial effects on the diet from multi-component, community-based intervention approaches. However, only a small number of earlier studies evaluated the effects of local community-based intervention sales based on data from local supermarkets. Using sales data provides the possibility to measure the effects of the intervention at the community level. Hence, although the target group of Project SoL was children aged 3–8 years old and their families, several of the intervention components were applied to the whole community (including interventions within the supermarkets and in the local mass media) and, thus, had the potential to promote healthy dietary changes among all citizens in the targeted communities. Furthermore, sales data provide a unique opportunity to objectively measure changes in diet-related behaviour. However, the intervention was most intensive among children and their families and, hence, the intervention effects might be considerably diluted when measuring effects at the community level instead of the individual level. Figure 2, Figure 3, Figure 4 and Figure 5 illustrate that the sales in all supermarkets varied during the intervention period, which is expected due to, for example, tourist seasons, which may explain the peak in sales in July 2013. The statistical analyses took this seasonal variation in sales during public holidays into account. Furthermore, the development in sales of different food groups also varied markedly, which can be explained by seasonal variation but also the change in focus on different foods during the intervention.

In this study, we did not find any effect of the intervention on sales of sweets, unhealthy snacks and sugary beverages. This is possibly because the intervention primarily focused on promoting the intake of healthy foods, including encouraging children to taste new foods products, and less on reducing the intake of unhealthy food products and beverages. The lack of effects on sales of sweets and sugary beverages might be one of the reasons why the intervention had no effect on children’s Body Mass Index (BMI) [36]. Compared to previous studies, both the Romp and Chomp Study [32] and Shape Up Somerville [34] succeeded in reducing children’s intake of sugary beverages and also found a significant decrease in BMI [32,34].

Unlike most other studies, we were able to evaluate the effects of a single intervention component (mass media intervention) as part of a multi-component intervention. This was carried out by measuring food sales on the whole island of Bornholm where the mass media intervention was implemented (level 2) and comparing it to food sales, both in the control area (level 3) and in the three SoL communities (level 1). To our surprise, a significant effect of the mass media intervention only was found on the sales of vegetables, wholegrain products and dried fruit compared to the control area (level 3). The reason may be that the local TV station was a very active partner in the project and supported and covered the project activities very intensively. Project SoL also had its own programme in TV2-Bornholm (called ‘Bare lidt sundere’ [Just a little healthier]), which was broadcasted once every month. Additionally, the supersetting approach supported the development of new innovative initiatives and ways to work with children, families and local communities that were interesting for TV2-Bornholm to cover. Furthermore, it can be hypothesised that part of the multi-component intervention in the three SoL communities inspired (‘contamination’) other areas on Bornholm and influenced citizens on Bornholm more broadly. Not least, it is likely that the other Coop supermarkets were inspired by the successful initiatives in the Coop intervention supermarkets due to their close collaboration. Consequently, the differences between the level 1 and level 2 areas might have been diminished and might also explain the small difference in sales data between the SoL supermarkets and ‘Bornholm’.

A strength of this study is the huge amount of data on food sales included from both the intervention areas and the control area. All supermarkets in both of the targeted municipalities (Bornholm and Odsherred) were involved in the intervention and data were collected during a period of 17 months. However, an important limitation of the study is that we were unable to include data from all these supermarkets. Two discount supermarkets (Netto) were located in the SoL communities and they actively participated in the SoL intervention. However, due to a shift in the data system in Netto during the intervention period, we were unable to include these data in the effect analyses. Furthermore, a small supermarket (Spar) was included in the study, but it was unable to provide digital sales data for the study. Therefore, we only included data from the Coop supermarket chain. Coop is, however, the largest supermarket chain in Denmark and there was a Coop supermarket in all three SoL communities (level 1). Furthermore, including data from the same supermarket chain strengthens the comparability of data between intervention and control supermarkets. The intervention in Netto and Spar followed the same overall themes and the same intervention strategies were used. Therefore, we have no reason to believe that their sales data differed from Coop.

A further limitation of the study is that the sales data from the first four months of the intervention (September–December 2012) were erroneously deleted by the supermarket chain. Therefore, the change in sales might be greater and, hence, the intervention effect larger than it was possible to measure because we would expect the greatest change at the beginning of the intervention period.

Overall, as mentioned earlier, the possibility of using sales data in the evaluation of an intervention provides a unique opportunity to objectively measure changes in diet-related behaviour. However, it is not possible to measure the changes in individuals or families included in the intervention. Furthermore, sales data are only a proxy of dietary intake and, therefore, sales data should be combined with measures of individual dietary intake, which was also undertaken in Project SoL [28].

The quasi-experimental study design with a matched intervention and control area has limitations with respect to the internal validity of the results [36]. However, complex intervention studies are often not suitable for an individual RCT study design. Future research should test the SoL intervention on a larger scale, preferably in a cluster-randomised design [37]. Furthermore, the target group of the intervention could be broadened to include more citizens and possibly the whole community.

## 5. Conclusions

The results showed significant increases in the sales of healthy food items in the supermarkets located in the high-intensity intervention area (level 1) compared to the low-intensity area (level 2) as well as the control area (level 3). Even the low-intensity area (level 2) showed a significant increase in sales of healthy food items compared to the control area (level 3). In contrast, there were no significant reductions in the sales of unhealthy food items, though this was not an explicit aim of the intervention. The study findings support the rationale for multi-component as well as singular mass media interventions as part of community-based health promotion initiatives.

## Figures and Tables

**Figure 1 ijerph-20-02478-f001:**
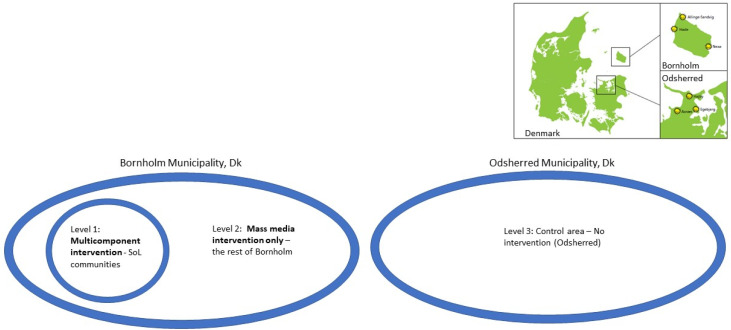
Evaluation design: comparing sales data for supermarkets in areas with a multi-component intervention (SoL communities), areas with mass media intervention only (the rest of Bornholm municipality) and the control area (no intervention, Odsherred municipality).

**Figure 2 ijerph-20-02478-f002:**
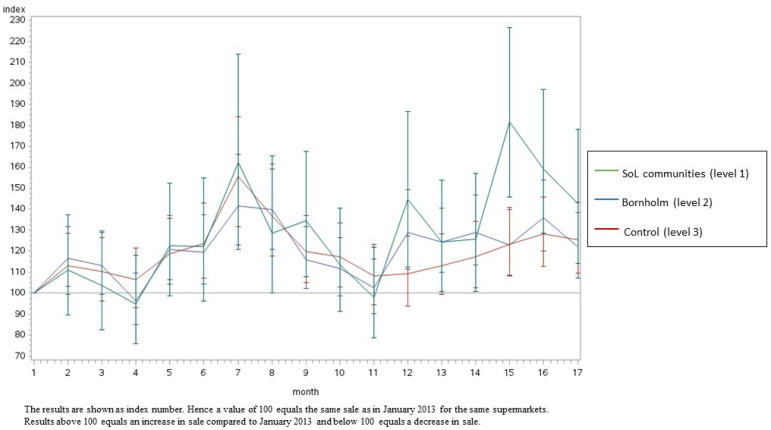
The development in monthly sales of fish (total) from January 2013 to June 2014 in the SoL intervention communities, the Island of Bornholm and the control area (Odsherred). Data are shown as index numbers.

**Figure 3 ijerph-20-02478-f003:**
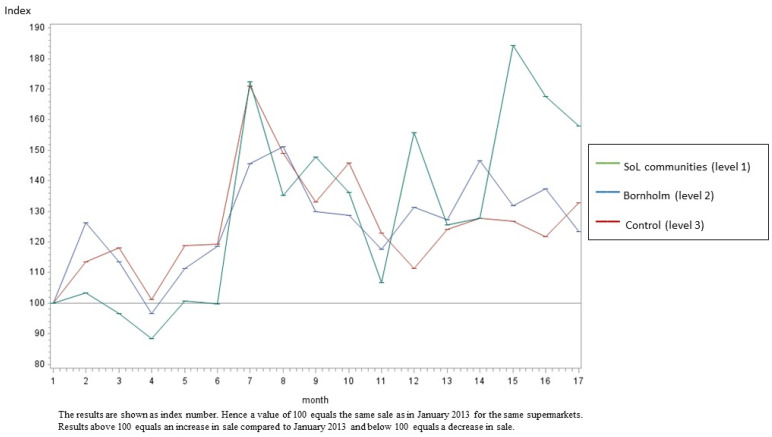
The development in monthly sales of canned fish from January 2013 to June 2014 in the SoL intervention communities, the Island of Bornholm and the control area (Odsherred). Data are shown as index numbers.

**Figure 4 ijerph-20-02478-f004:**
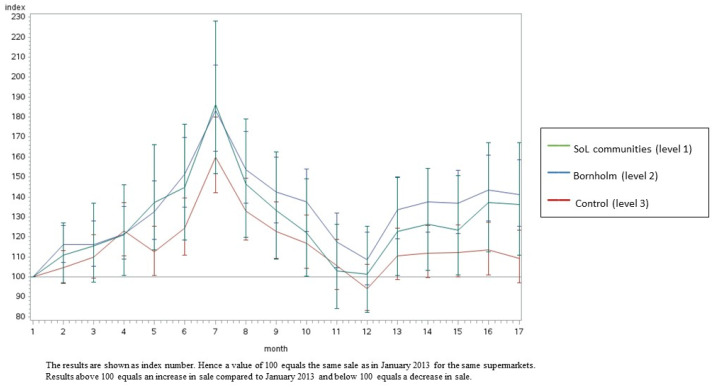
The development in monthly sales of vegetables (total) from January 2013 to June 2014 in the SoL intervention communities, the Island of Bornholm and the control area (Odsherred). Data are shown as index numbers.

**Figure 5 ijerph-20-02478-f005:**
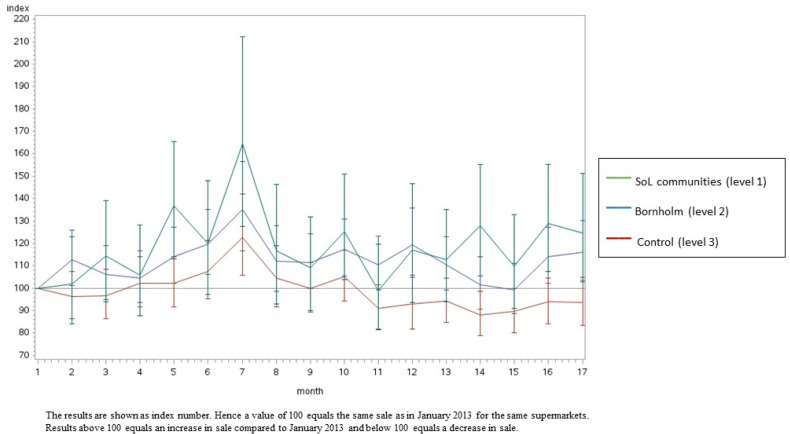
The development in monthly sales of oatmeal from January 2013 to June 2014 in the SoL intervention communities, the Island of Bornholm and the control area (Odsherred). Data are shown as index numbers.

**Table 1 ijerph-20-02478-t001:** Characteristics of the populations (>16 years) in Bornholm and Odsherred municipalities.

Category	Characteristic	Unit	Bornholm	Odsherred
**Population and area**	Population	1000	41	32
	Area, square km	Km^2^	588	355
**Health status**	Overweight, BMI > 25	%	50	53
	Diabetes	%	6.5	5.7
	High blood pressure	%	16	23
**Health behaviour**	Citizens with very unhealthy dietary habits	%	14	16
	Citizens with < 30 min/day MVPA *	%	36	41
	Citizens with self-perceived poor health	%	18	21
**Socio-economic status (SES)**	Unemployed	%	26	28
	No vocational education	%	19	18

* MVPA: moderate and vigorous physical activity; source: Poulsen et al. [25]; Glümer et al. [26]).

**Table 2 ijerph-20-02478-t002:** Effect of the intervention: comparing the development in the Coop SoL supermarkets (multi-component intervention, level 1), other Coop supermarkets on Bornholm (mass media intervention only, level 2) and Coop supermarkets in the control area (Odsherred municipality, no intervention, level 3).

	Multi-Component Intervention Vs. No Intervention (Control) Index		Multi-Component Intervention vs. Mass Media Intervention Only Index		Mass MEDIA intervention Only vs. No Intervention (Control) Index	
	Estimate	*p*-Value	Estimate	*p*-Value	Estimate	*p*-Value
Fish, total	**1.2950**	**0.0028**	1.1328	0.1565	1.1432	0.0659
Fish, canned	**1.3072**	**0.0253**	**1.3042**	**0.0249**	1.0023	0.9788
Vegetables, total	1.220	0.0595	1.0449	0.6761	**1.1675**	**0.0383**
Vegetables, fresh	1.1180	0.2518	0.9353	0.4775	**1.1976**	**0.0097**
Fruit, fresh	1.1135	0.4514	1.0448	0.7595	1.0658	0.5332
Fruit, dried	1.6004	0.0620	1.057	0.8225	**1.5138**	**0.0217**
Oatmeal	**1.3102**	**0.0028**	1.1041	0.2715	**1.1867**	**0.0080**
Wholegrain pasta	1.143	0.4727	0.7227	0.0834	**1.5815**	**0.0007**
Sweets	0.9876	0.8934	1.0116	0.9009	0.9762	0.7192
Sugary beverages	1.0620	0.5039	1.0012	0.9891	1.0608	0.3599

Results showing the effect of the intervention on the development in sales for selected food categories from development from the first month (January 2013) to the last three months (April, May and June 2014). The statistical analyses were adjusted for seasonal variation in sales during public holidays.

## Data Availability

The data presented in this study are available on request from the corresponding author. The data are not publicly available due to privacy.

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
