# Peer review of "A Community-Based, Participatory, Multi-Component Intervention Increased Sales of Healthy Foods in Local Supermarkets—The Health and Local Community Project (SoL)"

_ijerph, 2023, doi:10.3390/ijerph20032478_

Round 1

Reviewer 1 Report

Thank you for sharing your manuscript. I enjoyed learning more about your study. This is important work and difficult from a research perspective! But, parts of the methods and results needed more explanation.  For example, there was not much at all for theoretical or empirical foundations of this intervention, though I'm sure that the intervention design was based on theory and prior research. I also had trouble following along with the two intervention conditions (level 1 vs. 2) and settings between methods and results. It was hard to picture what was in the multi-component intervention and in which communities vs. the mass media campaign only and where that happened (all on an island). Consistency in language would be helpful (e.g., high-intensity intervention for level 1 vs. low-intensity intervention for level 2). I believe some rewriting will make it much easier for readers to follow your research.

ABSTRACT

Please revise after making other edits to sections and check that framing is consistent with presentation in manuscript.

INTRODUCTION

Overall - please revise to use more relevant framing, describe supersetting approaches in more detail, so it's easier for reader to understand rationale and significance of this study.

-I didn't follow the framing around family obesity for a food retail intervention with mass media campaign. Please consider rewriting to focus argument on Policy, System, and Environmental Strategies for food retail places, including media campaigns and supermarkets.

-I also didn't understand citing the U.S. dietary based guidelines document for opening sentence about role of diet in preventing diet-related chronic diseases and premature mortality for a Danish study. Please consider citing more relevant sources: something from the UN/WHO or EU, Danish studies based on national surveillance and monitoring, position papers from leading health/nutrition organizations, or systematic literature reviews and meta-analyses.

-Is there an acronym definition or explanation for the Danish Project SoL? If so please include it at first mention. If there's not, please don't worry.

-Is there a rationale and design paper for Project SoL? or a main effects paper? If so, please include citation(s). If there's not, then please eliminate the term "sub-study" if there has been no main effects paper. It's not clear to me if this paper is reporting on the main effects of Project SoL or secondary effects, based on what's presented in introduction. Based on description, Project SoL may have had multiple primary outcomes: local community resources, local social networks, physical activity, healthier eating and shopping behavior. (After I reviewed references, ref #27, #28, and #36 is for SoLPlease incorporate these papers into introduction.)

-I was not familiar with the  term "supersetting" before this peer review. Thanks for defining with a reference.

-Have others applied the supersetting approach? in Danish communities? or internationally? I know there have been intensive community-wide nutrition and physical activity interventions in the U.S. (e.g., Shape Up Somerville), but I'm not sure if they considered sustainability, so I'm not sure they would be considered part of supersetting. If there have been other examples in Europe or elsewhere, then please include those in the introduction.

I did a Google Scholar search and found these. If you don't know of his work, Joel Gittlesohn is a highly-respected, community-based interventionist who works with underserved communities in the U.S. I have learned a lot from this work and am sharing in case it's helpful. Please consider if they are relevant here:

Gittelsohn, J., & Trude, A. (2017). Diabetes and obesity prevention: changing the food environment in low-income settings. Nutrition Reviews, 75(suppl_1), 62-69.

Gittelsohn, J., Jock, B., Redmond, L., Fleischhacker, S., Eckmann, T., Bleich, S. N., ... & Caballero, B. (2017). OPREVENT2: Design of a multi-institutional intervention for obesity control and prevention for American Indian adults. BMC Public Health, 17(1), 1-9.

Gittelsohn, J., Novotny, R., Trude, A. C. B., Butel, J., & Mikkelsen, B. E. (2019). Challenges and lessons learned from multi-level multi-component interventions to prevent and reduce childhood obesity. International Journal of Environmental Research and Public Health, 16(1), 30.  

 Lehnert, T., Sonntag, D., Konnopka, A., Riedel‐Heller, S., & König, H. H. (2012). The long‐term cost‐effectiveness of obesity prevention interventions: systematic literature review. Obesity Reviews, 13(6), 537-553.

Another expert who may be helpful here is Christina Economos. Here are some of her papers:

Hatfield, D. P., Sliwa, S. A., Folta, S. C., Economos, C. D., & Goldberg, J. P. (2017). The critical role of communications in a multilevel obesity-prevention intervention: lessons learned for alcohol educators. Patient Education and Counseling, 100, S3-S10.

Hennessy, E., Korn, A. R., & Economos, C. D. (2019). A Community-Level Perspective for Childhood Obesity Prevention. In Global Perspectives on Childhood Obesity (pp. 287-298). Academic Press.

*The Community Guide by the CDC and this article talking about community-wide interventions that references the Community Guide:

Foltz, J. L., May, A. L., Belay, B., Nihiser, A. J., Dooyema, C. A., & Blanck, H. M. (2012). Population-level intervention strategies and examples for obesity prevention in children. Annual Review of Nutrition, 32, 391-415.

-As someone not familiar with Project SoL, it would been helpful to know a little more about the project at first mention, given the rather detailed aims of this manuscript. Please consider adding a reference in the introduction to let the reader know you'll be providing more detail in methods: "Project SoL targeted schools, childcare centres, and supermarkets as well as the local mass media, and social media (more detail on Project SoL in Methods section)."

-Please provide brief description of multi-component, so the reader can start to understand difference between multi-component and mass media campaign only in first two intervention groups.

METHODS

Please clarify evaluation/data collection to ensure that methods were presented clearly for an evaluation of supersetting intervention. The methods text specific to the mass media intervention and evaluation of the intervention did not seem sufficient.

There were some things I wasn't sure about:

Was the comparison community Odsherred also considered to be rural?

The intervention communities (receiving level 1 or level 2 of intervention) were all rural communities in Borholm. Please clarify the intervention setting and sample. In the manuscript, the setting is described differently (e.g., three rural communities, local schools and child care centers, supermarkets.)

If the levels 1 and level 2 of the intervention happened in the same place (three rural communities on island of Bornholm), how did the mass media intervention only affect some places? I was confused about how to consider a mass media intervention for an island.

How was dose considered? for level 1 or level 2 of the intervention?

It's difficult to understand the details of the mass media campaign from the text provided (three sentences). What were the theoretical or empirical foundations for the mass media campaign? Did the campaign change over the 19 months? Why was this 19-month period from January 2013 to June 2014 selected? When did the mass media campaign happen during the 4-year intervention? If the baseline Project SoL paper reported this already, then a brief description would be fine.

 It's difficult to understand if data were collected from all or most of the food retail places used by participants in the intervention (level 1 and level 2) and control (level 3) communities without more data to explain this. On p. 6 (lines 138-142), a flowchart would be helpful to show which food retail places participated, were exposed to intervention/treatment conditions, and which ones provided data for evaluation. This seems like an important missing piece.

I'm not sure if it's enough to evaluate intervention based on sales data from Coop supermarkets for a limited time period. Were there other supermarkets that were not Coop supermarkets, aside from the Netto and Spar?

Please make sure that results and discussion reflect data from the Coop supermarkets starting after month four.

In addition, sales data are typically very messy.

There was very little information provided on the sales data and how variables were calculated.

I'm not sure about the index used to determine change in food items sold in intervention period and there were no reference to help me understand this decision.

I do not have the expertise to evaluate the analytic approach for a study design like this and do not have any specific comments on appropriateness or things to improve. I'm trusting that the authors analyzed the data appropriately.

RESULTS

Please clarify results so that the change includes the time period. Is the change during the intervention period? or from before the intervention to after? Because this was a long intervention (4 years), with a relatively long mass media campaign (19 months?), it's hard to know to which time period this change refers. The figure titles include the month and year, which helps, but I wasn't sure where the 4-months of missing data were missing when looking at the figures.

Table 2 - difficult to interpret. Are data purchases/sales by food item? If so, please make this more clear in title and in table columns/headers.

Figures 2, 3, & 4 - Please add labels to show intervention communities (level 1 and level 2) so it's easier to interpret the figures. Please consider a different way to plot the data. It's hard to distinguish between the blue line and the green line.

I follow the decision to show figures for the biggest changes: total fish, oatmeal, and vegetables, but why not include a figure for canned fish, too?

For the figures, there are some places where the level 2 intervention had higher monthly sales of total fish vs. level 2 intervention (Figure 2 and Figure 3, first four or five months and at other points). The data for oatmeal seemed more consistent. Please describe this in results and help explain in discussion. 

DISCUSSION

Please revise to strengthen discussion after making updates to methods and results.

I wasn't expecting focus on dried fruit in the discussion when the results didn't really focus on fruit, but on canned fish, total fish, oatmeal, and vegetables. The opening paragraph in discussion highlighted findings from Table 2, but didn't seem to highlight findings from Figures 2-4.

Please re-read discussion and revise sentences that aren't supported (e.g., lines 270-271). There was little description of data collected re: consumer behavior for evaluation.

I also didn't understand period of 17 months, when earlier in paper I recalled period of 19 months.

The strengths and limitations would benefit from development and bringing in citations as needed. Please think about threats to internal validity and how with quasi-experimental study design, the potential influence of confounding, in addition to issues with sales data and issues w/using sales data as proxy for dietary intake.

Please integrate more data/findings from previous studies, including descriptions of settings/samples/intervention approaches and citations.

REFERENCES

Please update as you update the other sections and pay attention to formatting of bibliography.

-The number of references is quite efficient for this type of paper (36 references total). I believe that more references will be needed in some sections, but that additional references will only strengthen paper.

-The quality of the journals is good and authors of articles are experts in the field of childhood obesity and community-wide obesity interventions (like Shape Up Somerville).

-The formatting for references needs attention. Please check all references and ensure that complete reference information is provided. In addition, please verify correct format for articles, reports, book sections, and websites per author's instructions.

Correct formatting for journal articles is: Author 1, A.B.; Author 2, C.D. Title of the article. Abbreviated Journal Name Year, Volume, page range. As an example, in ref #2, article needs to be sentence case not book case; abbreviated journal name needs to be italicized and does not have punctuation; the period after journal name needs to be removed, etc. Correct formatting would be: Birch, L.L.; Anzman, S.L. Learning to eat in an obesogenic environment: A developmental systems perspective on childhood obesity. Child Dev Perspect 2010, 4, 138–143.

-Please consider including DOI for all journal articles. Ref #8 includes DOI though the formatting is incorrect (no hyperlinked text in references.)

-Please check that all references are in black text. There were some reference numbers in blue text.

Reviewer 2 Report

Very interesting and clear design intervention.

Follow below some of my suggestions and feedback:

Page 3, Line 46 – add “among adults”

Page 3 Line 52 – add “added sugars”

Page 3 Line 52 – which type of fat?

Page 3, Line 55 – rephrase for something like “Children and their families eating choices are part of a complex system including….”

Page 3, Line 72 – typo centers

Table 1 – what is considered a very unhealthy dietary habit, and how was it measured?

Table 1 – SES did you measure income? What is considered vocational education?

Page 5, Line 119 – typo centers

Page 5, Line 130 – typo centers

Page 6, Line 136 – what were the age of children from local schools and childcare centers?

Page 7, Line 195 – typo changes

What are the future recommendations for interventions similar to yours? Would it be interesting to expand beyond the children-parent dyad?

Also, what would you recommend as future tentative to decrease consumption of added sugars and saturated and trans fats?
